# Sub-minute prediction of brain temperature based on sleep–wake state in the mouse

**Yaniv Sela[1]\*, Marieke MB Hoekstra[2†], Paul Franken[2]\***

[1]Sagol School of Neuroscience, Tel Aviv University, Tel Aviv, Israel; [2]Center for Integrative Genomics, University of Lausanne, Lausanne, Switzerland

**Abstract** Although brain temperature has neurobiological and clinical importance, it remains unclear which factors contribute to its daily dynamics and to what extent. Using a statistical approach, we previously demonstrated that hourly brain temperature values co-varied strongly with time spent awake (Hoekstra et al., 2019). Here we develop and make available a mathematical tool to simulate and predict cortical temperature in mice based on a 4-s sleep–wake sequence. Our model estimated cortical temperature with remarkable precision and accounted for 91% of the variance based on three factors: sleep–wake sequence, time-of-day ('circadian'), and a novel 'prior wake prevalence' factor, contributing with 74%, 9%, and 43%, respectively (including shared variance). We applied these optimized parameters to an independent cohort of mice and predicted cortical temperature with similar accuracy. This model confirms the profound influence of sleep–wake state on brain temperature, and can be harnessed to differentiate between thermoregulatory and sleep–wake-driven effects in experiments affecting both.

**\*For correspondence:**
YanivDoar@gmail.com (YS);
paul.franken@unil.ch (PF)

**Present address:** [†]UK Dementia Research Institute at Imperial College London, Department of Brain Sciences, London, United Kingdom

**Competing interests:** The authors declare that no competing interests exist.

## Introduction

Brain temperature is a fundamental physiological variable that can affect numerous neural processes, from basic properties such as nerve conduction velocity, passive membrane potential, and synaptic transmission, to global regulation of brain activity (*Wang et al., 2014*). Specifically, fluctuations within the physiological temperature range (33–37°C in rodents) have been shown to modify channel kinetics (*Rosen, 2001*), reuptake of neurotransmitters by transporters (*Xie et al., 2000*), miniature postsynaptic currents (*Simkus and Stricker, 2002*), the neuronal firing rates of single neurons (*Tryba and Ramirez, 2004*; *Guatteo et al., 2005*), and neuronal synchronization (*Csernai et al., 2019*). Conversely, neuronal activity is one of the main determinants of brain temperature (*Kiyatkin et al., 2002*). Thus, brain activity both affects and is affected by fluctuations in temperature. In clinical settings, brain temperature increases in common pathological conditions such as stroke or head injury (*Mrozek et al., 2012*), and temperature is deliberately lowered in interventions to protect the brain from hypoxic events (*Faridar et al., 2011*). Since heat plays a crucial role in neuronal functioning (*Kiyatkin, 2010*; *Alonso and Marder, 2020*) and brain tissue is very sensitive to thermal damage (*Yarmolenko et al., 2011*), it is crucial to understand which factors contribute to changes in brain temperature under normal circumstances.

Generally, brain temperature is considered to be the net result of heat production, which is determined by brain metabolism, and heat dissipation, which is determined by brain blood flow and the brain-to-blood temperature gradient (*Hayward and Baker, 1969*). The sleep–wake states, non–rapid-eye-movement (NREM) sleep, rapid-eye-movement (REM) sleep, and wakefulness, define brain states associated with specific neuronal activities, oxygen consumption, and metabolism (*Nir et al., 2013*). The latter two (active) brain states are accompanied by increases in brain temperature, whereas NREM sleep evidences a decrease (*Hayward and Baker, 1969*; *Obál et al., 1985*;

*Franken et al., 1992a*; *Hoekstra et al., 2019*). Besides decreased heat production, increased heat dissipation further contributes to the decreases in brain temperature during NREM sleep: (i) although the global blood flow to the brain decreases in absolute terms, it increases when taking the considerable drop in oxygen consumption during NREM sleep into account (*McAvoy et al., 2019*), and (ii) since body temperature is actively down-regulated through peripheral vasodilation (and perspiration in humans) after sleep onset, the brain-to-blood temperature gradient increases (*Szymusiak, 2018*).

By quantifying the relationship between sleep–wake state and brain temperature we reported that the sleep–wake distribution explains 84% of the variance in brain temperature in the rat (*Franken et al., 1992b*), a finding we recently replicated in the mouse (*Hoekstra et al., 2019*). However, the 1992 analysis has been criticized for overestimating the impact of sleep–wake state by having averaged over hourly intervals and ignoring other contributing factors such as locomotion (*Heller et al., 2011*), and the sequential nature of assessing the contribution of sleep–wake and circadian-related factors (*Witting and Mirmiran, 1997*). Although recent studies have found the contribution of locomotor activity to brain temperature to be negligible (*Shirey et al., 2015*; *Hoekstra et al., 2019*) and confirmed that circadian factors do not contribute significantly to brain temperature (*Baker et al., 2005*), the concerns underlying the use of hourly values and a fixed order of assessing the contributing factors have not been directly addressed.

To resolve these outstanding issues, we drew on a recent dataset (*Hoekstra et al., 2019*) to develop a mathematical model that simulates changes in brain temperature at the high time resolution required to account for its rapid fluctuations during sleep–wake state transitions. This new model explained 91% of the variance in brain temperature and reduced the model error to 0.26°C down from an observed dynamic range of 3.13°C. In addition to accurately capturing the short-term dynamics associated with sleep–wake transitions, the model revealed prior wake prevalence as a novel, longer-term factor altering the range of values within which brain temperature is regulated. The circadian factor explained 9.3% of the overall variance in brain temperature, of which 7.6% was redundant with the contribution of the other two factors. Finally, we show that the model can accurately predict brain temperature dynamics in an independent cohort of mice using the parameter settings obtained in the main experiment. This model thus, contributes to better documenting and quantifying the fundamental dependence of brain temperature on the sleep–wake state.

## Results

Wakefulness and REM sleep can be considered to be the opposite of NREM sleep in brain temperature dynamics. In the cortical activated states of wakefulness and REM sleep, brain temperature increases, whereas during NREM sleep when cortical input is reduced and neuronal activity becomes synchronized, temperature decreases (*Obál et al., 1985*; *Franken et al., 1992a*). Therefore, based on a previously described model in the rat (*Franken et al., 1992b*), we used the following exponential equations to iteratively simulate changes of brain temperature in the mouse:

$$\text{During waking/REM sleep}: T_t = U - (U - T_{t-1}) \cdot e^{\frac{-\Delta t}{\tau_{WR}}}$$

$$\text{During NREM sleep}: T_t = L - (T_{t-1} - L) \cdot e^{\frac{-\Delta t}{\tau_N}}$$

With a time step ($\Delta t$) of 0.0011 hr (i.e., the 4-s epochs at which sleep–wake states were scored), the current temperature ($T_t$) was calculated based on the preceding temperature ($T_{t-1}$) according to the distance from an upper asymptote ($U$) and time constant $\tau_{WR}$ when the mouse was awake or in REM sleep at time $t$ or, when in NREM sleep, according to the distance from the lower asymptote ($L$) and time constant $\tau_N$. The four free parameters of this basic model were the two time constants (in hours) and the values of the two asymptotes (in °C) between which brain temperature could vary. Drawing on previous results and assumptions (*Franken et al., 1992b*), we initially set the values for both time constants to 0.47 hr, the lower and upper asymptotes to the minimum and maximum temperatures reached in each animal during the 96-hr recording, respectively, and the initial temperature ($T_0$) to the average temperature in the first 5 min of the recording. However, the following

features were further developed to improve the simulation: (1) as already mentioned above, we defined 2 different time constants to represent the increase and decrease in temperature, instead of just 1 for both processes, (2) the asymptote values were free parameters, (3) all free parameters were simultaneously optimized for each mouse, (4) optimization took into consideration the entire recording, including sleep deprivation and recovery, instead of the baseline period alone, and (5) performance was assessed at a sub-minute time scale; that is, the 4-s resolution at which the sleep–wake states were determined, rather than at an hourly resolution.

Based on this more refined model, referred to as Model 0, we simulated the temperature recordings based on the individual sleep–wake state sequence data (*Figure 1*), with an average correlation coefficient (r) of 0.91, and a median root mean square (RMS) error of 0.36°C across animals. Generally, the time constant of wakefulness was higher than the one of NREM sleep (i.e., in 9 out of 11 mice; with median of 0.33 vs. 0.23 hr), indicating that the rate of increase of cortical temperature during wakefulness was slower than its decrease in NREM sleep. The difference between the upper and lower asymptotes varied by roughly 3°C, with the upper asymptotes consistently near the temperature values observed during SD (*Supplementary file 1*).

However, the simulation presented deviations from the measured cortical temperature, especially when higher temperatures were reached during the light phase and sometimes around lower values during the dark phase (see arrows in *Figure 1A*). When we examined the time course of the average hourly residuals of the model across animals (*Figure 1B*), we found a systematic fluctuation in the fit; during the baseline light periods simulated values were too high and during the dark periods too low, compared to the recorded temperature values. The residual RMS (i.e., the RMS of the averaged hourly residuals) amounted to 0.19°C. However, the residuals during sleep deprivation (SD) which occurred during the light period resembled those observed during the dark period, which argues against a simple circadian modulation. Although incorporating a circadian modulation of both asymptotes as previously done in the rat (*Franken et al., 1992b*) somewhat improved the overall fit (RMS error = 0.32°C; residual RMS = 0.15°C), it did not abolish the apparent periodicity in the baseline residuals in mice and resulted in an even poorer fit during SD (*Figure 1—figure supplement 1*). Given the time-of-day independent similarity between the residuals during the SD and the dark phase, an alternative factor that could contribute to a temporary upregulation of brain temperature (beyond the sleep–wake state- driven changes already captured by the simulation) could be the prior periods of sustained wakefulness (*Obermeyer et al., 1991*). To explore this possibility, instead of a circadian modulation, we changed the asymptotes according to the prevalence of wakefulness (and REM sleep, for consistency) prior to each data point. We refer to this factor as 'prior wake prevalence'. The window size over which prior wake prevalence was calculated, as well as the time lag for affecting the asymptotes, were kept as free parameters. Modulation of the asymptotes according to prior wake prevalence considerably improved the fit (r = 0.95; RMS error = 0.28°C; *Figure 2*) and removed most of the excessive overestimations during light periods, as well as the underestimation of temperature during the dark periods (see arrows in *Figure 2A*). We found that the optimal window size was 4.0 hr, with a shift of 1.5 hr prior to the time point under consideration, and a scaling factor of 1.2°C. The latter parameter represents the maximum possible modulation of either asymptote (i.e., 100% wakefulness or 100% NREM sleep during a given 4.0 hr window; *Supplementary file 2*). We optimized all the parameters simultaneously in this new model (i.e., Model 1) and found that all of the values obtained with Model 0 changed significantly (p≤0.002; $F_{(2,10)} \geq 9.98$, one-way repeated measures analysis of variance [rANOVA]): the lower asymptote increased, leading to a substantial reduction in the inter-asymptote temperature range from 3.1°C to 2.0°C, which might have contributed to the substantial shortening of the time constants (0.18 and 0.13 hr for wake/REM sleep and NREM sleep, respectively), as they need to be faster to compensate for the reduced distance from the asymptote so that a similar increase/decrease temperature rate can be achieved.

Interestingly, after incorporating the prior wake-prevalence factor, the residuals of the new model showed a consistent light–dark (circadian) modulation (*Figure 2B*), with an over-estimation of the temperatures in the light periods, including the SD (residual RMS = 0.12°C). To account for this modulation, we again applied a 24 hr sine-wave modulation onto the asymptotes. The combined effect of modulating the asymptotes according to prior wake-prevalence and circadian time (i.e., Model 2) almost flattened the residuals (residual RMS = 0.09°C; *Figure 2C*) and further improved the fit (r = 0.96; RMS error = 0.26°C). The amplitude of the sine wave was 0.19°C with a phase of −0.63 hr,

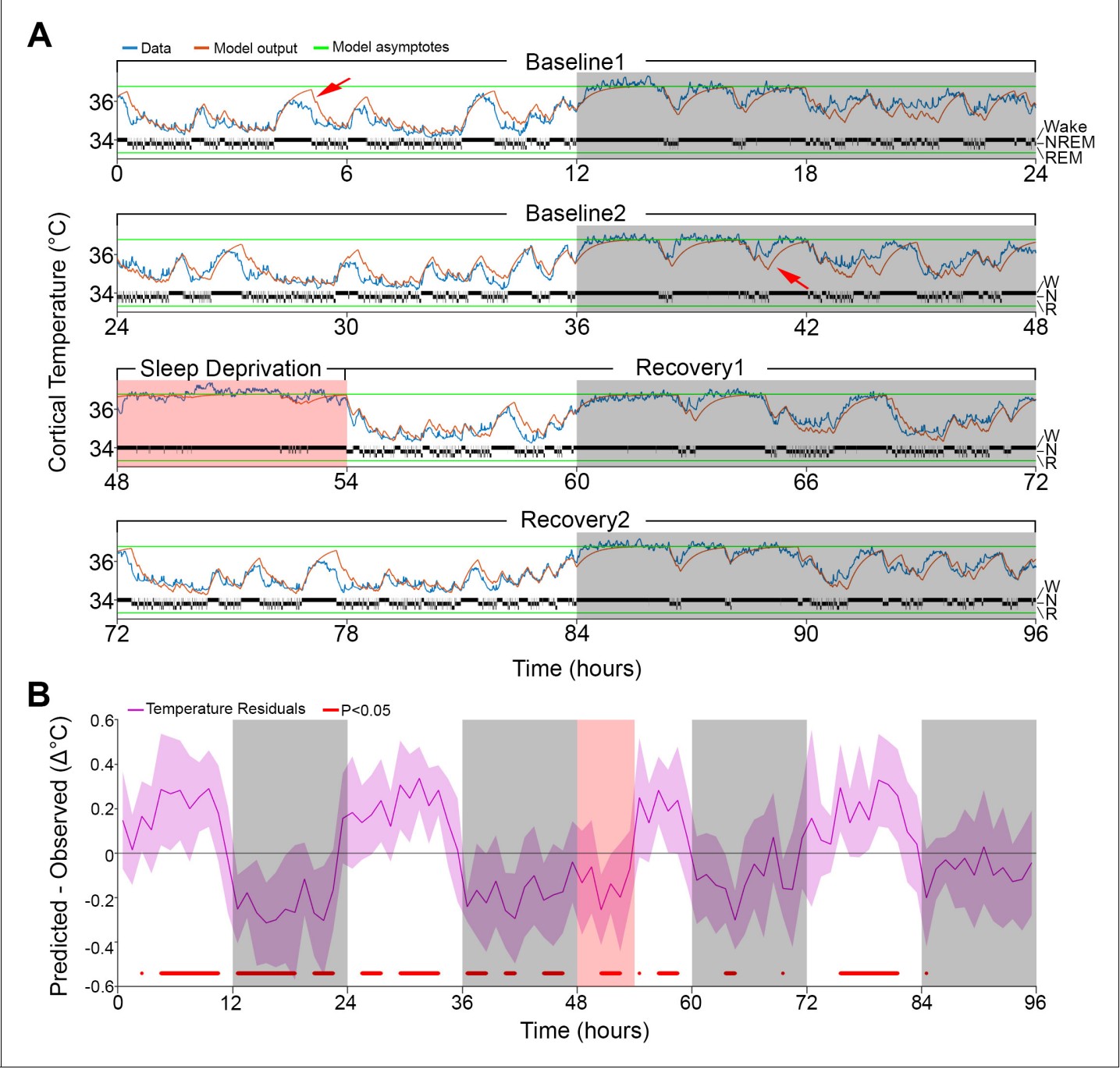

**Figure 1.** Results of Model 0 with constant asymptotes. (**A**) A representative example of a 96-hr recording in one mouse of brain temperature (blue) and simulated data (orange). Green lines represent the model's lower and upper asymptotes. The 4-s hypnogram of wake (W), non–rapid-eye-movement sleep (N), and rapid-eye-movement sleep (R) appears above the lower asymptote. White/gray backgrounds represent 12 hr light–dark periods, respectively, and the salmon background starting at 48 hr indicates the 6 hr of sleep deprivation. Red arrows point to examples of over/under estimation of the model in the light–dark periods, respectively. (**B**) Hourly differences (mean ± STD) between simulation output and data. Red marks below the graph represent significant differences, tested by paired $t$-tests and false discovery rates corrected at $p<0.05$. Hourly values are plotted at the interval midpoint. White/gray/salmon backgrounds as in (A).

The online version of this article includes the following figure supplement(s) for figure 1:

**Figure supplement 1.** Residuals of the model in which both asymptotes were modulated according to a circadian rhythm.

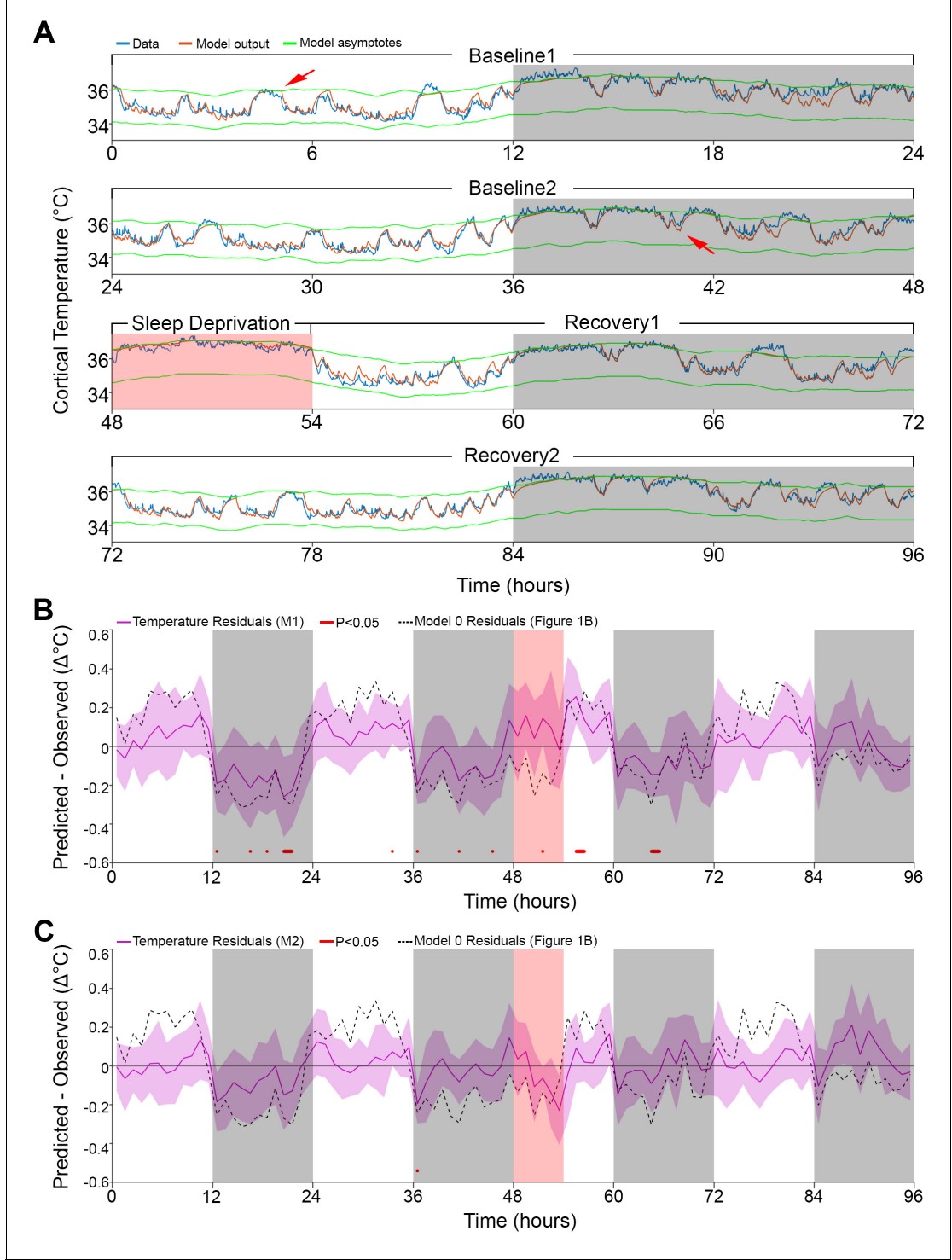

**Figure 2.** Results of models 1 and 2 with modulation of asymptotes. (**A**). Simulation fit (orange) after incorporating prior wake prevalence (Model 1). Note that both asymptotes (green) are modulated in parallel, recorded data (blue line) are of the same animal as in *Figure 1*, and previous over- and under-estimations marked by red arrows are diminished. White/gray/salmon backgrounds indicate light/dark/SD periods, respectively. (**B,C**). Residuals of models 1 and 2, as in *Figure 1B*, after the addition of prior wake prevalence (B, Model 1, M1) and with an additional circadian rhythm modulation of both asymptotes (C, Model 2, M2). Dashed lines mark the mean temperature residuals of Model 0, from *Figure 1B*. Red marks indicate significant deviations from zero. Note the reduction in the number of red marks from *Figure 1B*, to *Figure 2B* to *Figure 2C*.

placing the trough of the circadian influence at ZT5.37. The window size over which the prior wake-prevalence was calculated shortened to 3.0 hr and the difference between the asymptotes decreased to 1.85°C (*Table 1*). Relative to Model 1, the only free parameter that exhibited a significant change was the scaling factor of the prior wake prevalence window (which dropped from 1.2 to 1.0°C, paired *t*-test, $T(10) = 5.4$; p=0.0003). *Figure 3* shows the final fit of Model 2 to the temperature data for each of the 11 animals. Note that mouse #616 displayed exceptionally long waking periods (see periods of uninterrupted high temperature levels in the dark periods) that likely contributed to the exceptionally long prior wake prevalence window (5.5 hr) and the large scale factor relative to the circadian influence (1.05 vs. 0.11°C), modulating the asymptotes in this animal. These aberrant parameters, might in turn, explain the exceptional phase of the trough of the circadian factor (ZT15.6).

Since Model 2 explained roughly 91% of the variance in the data, we next examined the contribution of each of the three factors. To do so, we decomposed the simulated temperature signal into its three constituent factors by removing the circadian and/or prior wake-prevalence factors, and subtracting the respective result from the Model 2 output (see 'Materials and methods'). Consistent with the $R^2$ value of 83% achieved in Model 0, the factor sleep–wake state accounted for the largest portion of the variance, that is, 74% (*Figure 3—figure supplement 1*). Some of this explained variance was, however, shared with the other two factors in the model, that is, prior wake prevalence (27%) and circadian time (4%), leaving 42% as the unique contribution of the sleep–wake state. In comparison, the uniquely explained variance contributed by the prior wake prevalence and the circadian process were considerably smaller (12% and 2%, respectively). In total, the overall explained variance contributed by prior wake prevalence was 43% and by the circadian process 9%, of which 3% was shared.

In 5 of these 11 mice, we ran additional experiments with a similar design but shorter SD (2 and/or 4 hr, starting at ZT0) and only one day of recovery. To verify whether the parameters found in the main experiment were not overfitted to the specific experiment, we tested the performance of Model 2 in each of the additional recordings, using the individually optimized parameters from the 6

**Table 1.** Model 2 parameters for each animal.
Optimized values for each of the model parameters, and additional three descriptive variables: the difference between the upper and lower asymptotes (fourth column), the root mean squared error, and the correlation coefficient (two last columns). Columns 7–9 list the parameters of the prior wake-prevalence window: 'Size' refers to the window size, 'Shift' to the interval between the end of the window and the time point being evaluated, and 'Scale' to the conversion from % waking within the window to its temperature modulation of the asymptotes. Column 11 lists the phase of the 24-hr sinewave modulating the asymptotes (starting at zero) relative to ZT0. The last row summarizes the median value for each parameter, except for the correlation coefficient which is averaged after a Fisher transformation. Asterisks indicate KO mice.

| Animal | Asymptotes (°C) | | | Time constants (hr) | | Prior wake prevalence | | | Circadian | | RMS error (°C) | Correlation |
|---|---|---|---|---|---|---|---|---|---|---|---|---|
| | Lower | Upper | Difference | Wake/REM | NREM | Size (hr) | Shift (hr) | Scale (°C) | Amplitude (°C) | Phase (hr) | | |
| 603 | 34.26 | 35.82 | 1.56 | 0.21 | 0.06 | 4.75 | −1.90 | 1.02 | 0.13 | 0.86 | 0.28 | 0.94 |
| 606 | 34.74 | 36.59 | 1.85 | 0.23 | 0.10 | 2.75 | −1.30 | 1.01 | 0.25 | −0.71 | 0.26 | 0.95 |
| 608 | 32.28 | 33.77 | 1.49 | 0.22 | 0.08 | 2.75 | −1.40 | 0.92 | 0.25 | −0.41 | 0.23 | 0.95 |
| 609 | 34.04 | 36.28 | 2.24 | 0.22 | 0.11 | 1.75 | −1.10 | 0.68 | 0.19 | −2.66 | 0.25 | 0.97 |
| 612 | 32.60 | 34.45 | 1.85 | 0.23 | 0.11 | 1.50 | −0.90 | 0.73 | 0.18 | −0.63 | 0.24 | 0.97 |
| 613 | 34.15 | 36.07 | 1.92 | 0.11 | 0.17 | 3.25 | −1.00 | 1.12 | 0.26 | −0.83 | 0.30 | 0.93 |
| 616* | 34.01 | 36.11 | 2.10 | 0.17 | 0.14 | 5.50 | −2.70 | 1.05 | 0.11 | 9.59 | 0.26 | 0.97 |
| 617* | 34.54 | 36.30 | 1.76 | 0.16 | 0.16 | 3.25 | −1.00 | 1.01 | 0.17 | −1.17 | 0.23 | 0.96 |
| 619* | 34.53 | 36.40 | 1.87 | 0.21 | 0.11 | 3.00 | −1.60 | 0.62 | 0.06 | −1.07 | 0.25 | 0.96 |
| 620* | 36.89 | 38.21 | 1.32 | 0.12 | 0.05 | 2.75 | −1.40 | 1.16 | 0.22 | 1.29 | 0.29 | 0.95 |
| 622 | 36.14 | 38.11 | 1.97 | 0.20 | 0.09 | 4.75 | −2.00 | 1.02 | 0.19 | −0.41 | 0.31 | 0.95 |
| Median | 34.26 | 36.28 | 1.85 | 0.21 | 0.11 | 3.00 | −1.40 | 1.01 | 0.19 | −0.63 | 0.26 | 0.96 |

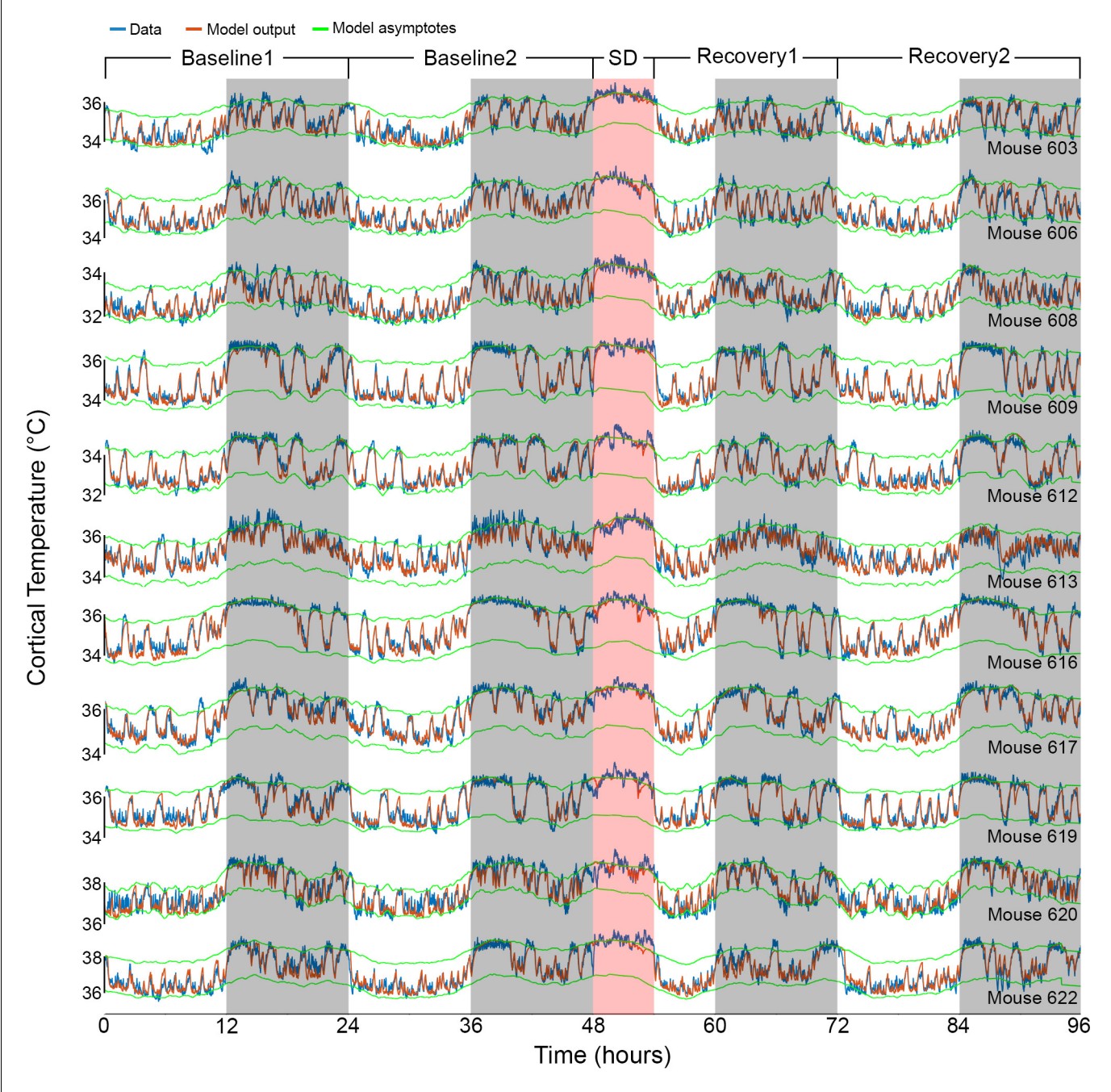

**Figure 3.** Results of Model 2 for all individual mice. The graphs show the fit (orange) of the final model to the data (blue) of all 11 animals (ordered as in *Table 1*). Example data in *Figures 1A* and *2A* correspond to mouse number 617 in the 8th row. White/gray/salmon backgrounds indicate light/dark/ sleep deprivation (SD) periods, respectively.

The online version of this article includes the following figure supplement(s) for figure 3:

**Figure supplement 1.** Proportional Venn diagram of the variance explained by each of the three factors in Model 2.

hr SD experiment (*Table 1*). The simulation proved to be robust for all the new recordings (all r values ≥0.91, and RMS errors ≤0.37), thus convincingly demonstrating the generalization of parameters within animals over time and across different experiments (*Supplementary file 3*).

Finally, given the high precision of the simulations, we inquired whether the algorithm could go beyond the simulations based on parameters adjusted specifically to an individual mouse. To this

end, we used the median parameter values found previously (*Table 1*) to predict brain temperature in a different cohort of mice (n = 5) recorded in the context of another study. The experiment had the same 96 hr design as the current study, and none of those data were used to optimize the model parameters. To test the model without providing temperature data, and because the model is iterative, we needed to estimate the initial temperature ($T_0$) to start the first iteration. We first used the main dataset to associate the average temperature and the percentage of Wake/REM state in the first 7 min of recording by linear regression, and then estimated the $T_0$ values in the current dataset based on wake/REM prevalence in the first 7 min (see 'Materials and methods', and *Figure 4—figure supplement 1*). After obtaining $T_0$, we used Model 2 to simulate the brain temperature of the independent cohort. Remarkably, all the correlation coefficients were between 0.93 and 0.95, although the median of the RMS error was 0.48°C since some recordings had consistent differences in absolute temperatures from physiological values, as was also observed in the main dataset (see columns 2-3 in *Table 1*). When we brought the empirical temperature data to the same average level as the predicted temperature traces (without changing scale), the median of the RMS error was reduced to 0.34°C and the fine overlap between predicted and observed temperature measures was again revealed (*Figure 4*, and *Figure 4—figure supplement 2*). Repeating this process for the test group with the medians of the more basic models 0 and 1, yielded correlation coefficients in the 0.86–0.89 and 0.91–0.93 ranges, respectively.

## Discussion

We developed and made available a tool to predict brain temperature dynamics based on the sleep–wake state sequence. The model showed a very accurate fit with data obtained under undisturbed baseline conditions, and during and following SD of varying lengths. It equally well predicted the global temperature dynamics on a time scale of hours and the changes following sleep–wake transitions on the order of seconds. In addition to two known factors modulating brain temperature, that is, the sleep–wake state and circadian influences, we identified a novel contributing factor involving the prior wake prevalence, which accounted for the up-regulation of brain temperature observed during periods of sustained wakefulness.

### Model parameters

In mice kept under our experimental conditions, specifically a 25°C ambient temperature, a 12:12 hr light–dark cycle, housed singly, and *food* ad libitum, we observed a dynamic brain temperature range of a little over 3°C. However, at any given time of the experiment, the range of observed temperature fluctuations did not surpass 2°C. This range, delimited by the upper and lower asymptotes in the model, represents a homeostatically defended range within which the brain temperature can vary according to the animal's behavior without eliciting a thermoregulatory response (*Parmeggiani et al., 1975*; *Satinoff, 1983*). Both asymptotes were modulated by two factors. The first was a time-of-day factor modeled by a sine wave, lowering and raising the defended temperature range with an amplitude of 0.19°C, reaching lowest levels close to the light period midpoint, that is, ZT5.4. Although we referred to this factor as 'circadian', given the experimental conditions, this fluctuation could also relate to the imposed light–dark cycle. Nevertheless, using a different analytical approach, our previous work in rats maintained under different photoperiods and under constant dark conditions arrived at very similar amplitudes for the time-of-day modulation of cortical temperature (0.13–0.21°C; *Franken et al., 1992b*; *Franken et al., 1995*), supporting the interpretation that the time-of-day factor does not depend on the lighting condition and is likely to represent a modulation of circadian origin.

The model identified a second factor dubbed 'prior wake-prevalence', that modulated the asymptotes of the defended temperature range. This factor differs considerably from the acute dependency of brain temperature on sleep–wake state transitions considered in Model 0 in that it integrates sleep–wake state information over several hours instead of minutes, thus suggesting a different underlying mechanism. Practically, it quantifies an up-regulation of the level at which temperature is regulated after sustained periods of wakefulness. Up-regulation of brain temperature during sustained wakefulness has been reported previously but was observed under conditions of SD spanning several weeks (*Obermeyer et al., 1991*). Because the prior wake-prevalence effect here was transient, and only involved a 3-hr time window, and was equally observed under baseline and

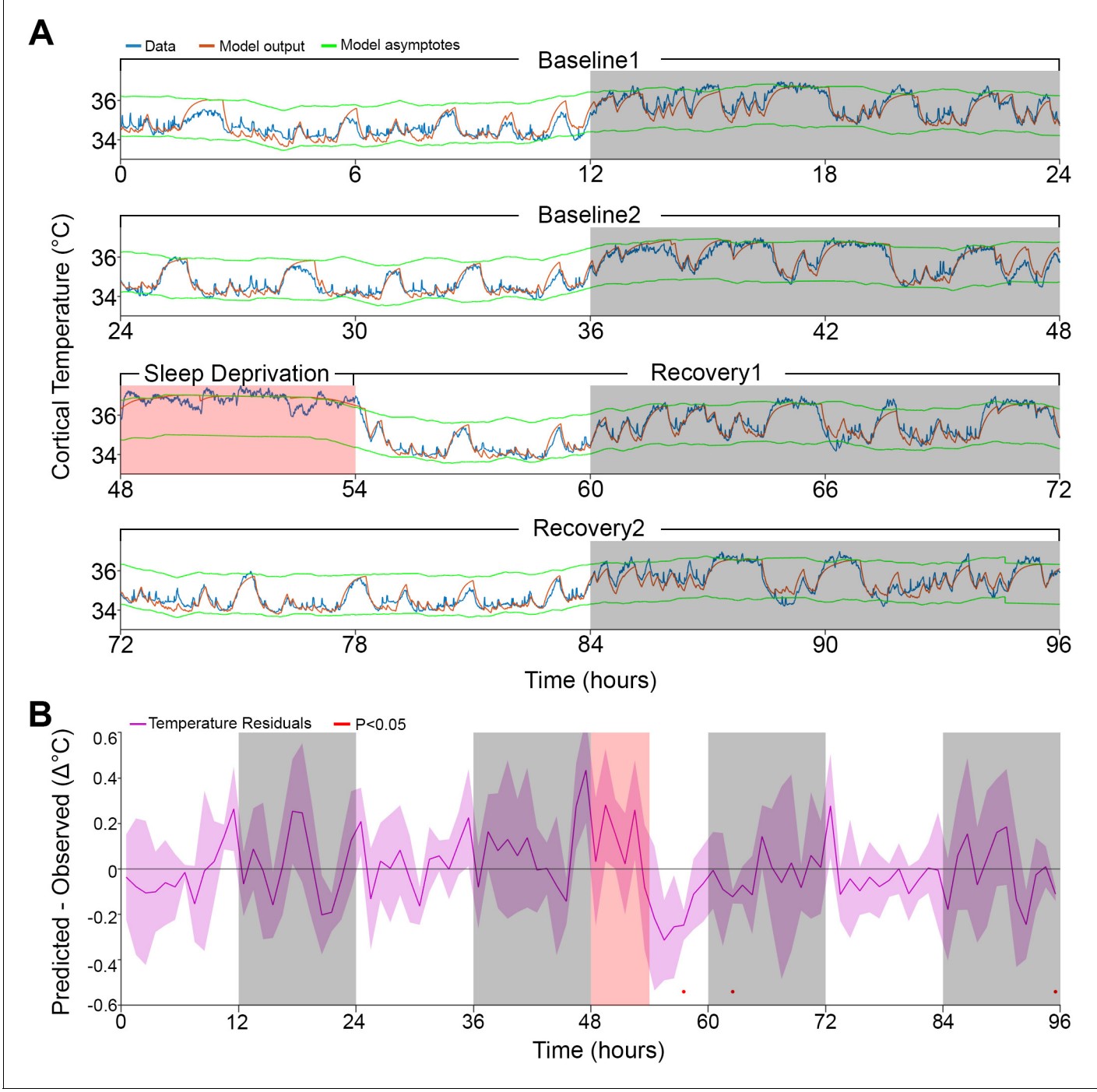

**Figure 4.** Model fit to a novel dataset. (A) Representative example of the Model 2 fit (orange) to novel raw data (blue) not used for optimization, using the median of the optimized parameters from the original dataset (*Table 1*). (B) Temperature residuals of Model 2 (mean ± STD) across all animals in the novel dataset. Notice the small number of red markers, indicating significant deviations from zero. White/gray/salmon backgrounds in both panels indicate light/dark/sleep deprivation periods, respectively.

The online version of this article includes the following figure supplement(s) for figure 4:

**Figure supplement 1.** Correlation between initial temperature to wake and rapid-eye-movement (REM) sleep prevalence.

**Figure supplement 2.** Results of Model 2 for each mouse of the independent cohort.

sleep-deprivation conditions, the underlying mechanisms might not be the same. The maximal amplitude of the modulation of the asymptotes brought about by the prior wake-prevalence factor (i.e., the scaling parameter in *Table 1*), was about fivefold larger than that of the circadian modulation (1.01°C vs. 0.19°C), thus underscoring the importance of prior wake prevalence in determining brain temperature. Interestingly, the prior wake-prevalence effect affected the asymptotes 1.4 hr later. Although we are unable to provide a satisfactory explanation for this finding, delayed effects of sleep–wake state transitions have been reported in humans for body temperature (*Bunnell et al., 1988*; *Youngstedt et al., 1997*), suggesting that the delay we observed in the mouse might relate to lagging peripheral effects affecting body temperature and, subsequently, brain temperature. Although we refer to this factor as prior wake-prevalence, it is equally plausible that the predominance of NREM sleep in a given interval drives the modulation. The onset of NREM sleep is associated with an active down-regulation of brain temperature involving peripheral vasodilation concomitant with decreased neural activity (*Glotzbach and Heller, 1976*). Therefore, the 3 hr time windows during which NREM sleep prevails might elicit a delayed net decrease in brain temperature involving peripheral mechanisms reminiscent of the slow physiological responses characteristic of sleep inertia in humans (*Kovac et al., 2020*). Our current data do not lend themselves to identify this factor's physiological substrate and needs further investigation.

The other optimized parameters were the time constants describing the changes in cortical temperature occurring at the sleep–wake transitions. Mathematically, a time constant value represents the time it will take to close ~63% of the gap between the current temperature and the asymptote, a duration that is constant regardless of the actual distance from the asymptote. To limit the number of free parameters in the model we did not differentiate between the rates of increase in wakefulness and REM sleep. Although this assumption is likely to be false (see, e.g., *Hoekstra et al., 2019*), we reasoned that it would not impact the model's performance significantly since REM sleep was relatively sparsely represented in our data (6% of recording time). The simulation identified time constants in the range of minutes for both the increases during wakefulness (and REM sleep) and the decreases in cortical temperature during NREM sleep. It also found longer time constants for the increase than for the decrease (13 and 7 min, respectively), pointing to a slower buildup of heat relative to its dissipation during NREM sleep. This difference is consistent with the concept that the blood that perfuses the brain acts as a heat sink, and that increases in temperature underestimate the rate of heat production since cooling occurs simultaneously (*Hayward and Baker, 1969*).

## Explained variance

Using linear regression analysis between the hourly values of wakefulness and brain temperature, the sleep–wake state was reported to explain 84% of the variance in brain temperature (*Franken et al., 1992b*; *Hoekstra et al., 2019*). Although the overall fraction of the variance explained in the final iteration of our simulation was higher (91% in Model 2), the sleep–wake state factor explained less of the variance in brain temperature (74%; see *Figure 3—figure supplement 1* for a summary). This discrepancy is due to the additional factors the model identified, primarily the prior wake-prevalence with which the sleep–wake state factor shared an important part of the explained variance. The linear regression results mentioned above should optimally be compared to the results of Model 0, in which only the sleep–wake state sequence was considered. In fact, Model 0 explained a very similar portion of the variance, that is, 83%. As suggested previously (*Franken et al., 1992b*), the circadian factor carried surprisingly little information about brain temperature. Of the 9% of the variance that could be attributed to circadian modulation, most was redundant with that of the other two factors and only <2% was uniquely circadian. In the model, both the circadian and prior wake-prevalence factors modulated the asymptotes. Consistent with the fivefold larger portion of variance explained by the latter, the maximum possible modulation of the asymptote was also fivefold larger.

## Comparison to previous models

Compared to our earlier effort to simulate brain temperature in the rat (*Franken et al., 1992b*), we made a number of key modifications that enhanced the accuracy of the simulation. In the previous model, only the time constants were optimized but the estimates for the asymptotes were taken directly from the data. Moreover, the same time constants were used to simulate the increases and

decreases in temperature accompanying sleep–wake state transitions. The current analysis showed that increase and decay rates differ considerably with a faster temperature decay in NREM sleep than the increase in wakefulness. Other factors estimated from the residuals in the original model were not formally optimized in one model simultaneously. The first such factor is the time-of-day ('circadian') modulation, whose amplitude was surprisingly similar in the two studies and species. Moreover, in the rat study, a down-regulation of the asymptote was needed to accurately fit the initial 12 hr of recovery because the actual temperature levels were lower than those predicted based on the sleep–wake distribution. This down-regulation after SD was not observed in the current study even when the prior wake-prevalence factor was not incorporated (*Figure 1—figure supplement 1*). This difference could reflect a thermoregulatory response to the longer SD in the rat (24 hr vs. 6 hr in the mouse) or could result from the assumption of equal decay and increase rates, because during the recovery, time spent in NREM sleep was increased and in the current study NREM sleep was found to be associated with a faster decay rate. The final difference between the two studies is the considerably slower time constants in the rat (0.47 hr) compared to the mouse, which may be attributed to a species difference.

The differences between the simulations were probably also affected by the modifications we made to address previous criticism. As previously noted by others (e.g., *Witting and Mirmiran, 1997*), the sequential evaluation of the contribution of different factors may lead to an overestimation of those assessed first because shared (or redundant) variance will be added to the first but not the second. We addressed this problem by optimizing the parameters for the various factors simultaneously and by explicitly assessing the shared variance among factors. Another criticism concerned the use of hourly mean values to evaluate the model's performance, since it removes a large portion of the variability and leads to inflated correlations (*Heller et al., 2011*). We found no support for this criticism: performance assessed at a 4 s resolution yielded high correlations similar to the hourly values (*Franken et al., 1992b*).

## Additional factors affecting brain temperature dynamics

Our simulation predicted brain temperature with high accuracy under the specific conditions of our study, but other factors affecting brain temperature could not be considered. For example, the efficacy with which blood removes heat from the brain depends on the ambient temperature (*Hayward and Baker, 1969*), which was kept constant during experiments. Although brain temperature dynamics during sleep–wake state transitions seem qualitatively similar across a wide range of ambient temperatures (*Alföldi et al., 1990*), it is nevertheless likely that the model parameters, and in particular the time constants, would need to be optimized for each ambient temperature. Moreover, increases in brain temperature accompanying intense activity such as wheel running (*Fuller et al., 1998*; *Kunstetter et al., 2014*), or activating stimuli such as a tail pinch or cage change (*Kiyatkin et al., 2002*), cannot be captured by the current version of our model because we considered wakefulness to be a uniform brain state. Similarly, sleeping alone or in a group, or having access to a nest, will likely affect the dynamics of brain temperature during sleep through its impact on heat dissipation (*Gordon, 2017*). Although these refinements of the simulation are currently lacking, the model could be easily expanded to accommodate these factors, once experimental data are available.

## Brain vs. body temperature

Rhythms in temperature are generally considered a direct output of the circadian time-keeping system (*Refinetti and Menaker, 1992*). Our model showed, however, that the circadian contribution to brain temperature in the mouse is small and that the observed circadian rhythmicity in brain temperature under undisturbed conditions is driven primarily by the circadian sleep–wake distribution. Because circadian studies often rely on core body temperature measures, it remains unclear whether the model could predict changes in body temperature with similar precision. In rodents in general, body and brain temperatures change in parallel (*Blessing, 2018*), suggesting that their gross dynamics are governed by similar rules and that the daily dynamics (e.g., hourly mean values) of body temperature can be predicted using assumptions similar to those in our model. However, at the finer time scale at which we optimized our model, the few available studies suggest that the temperatures of the brain and body behave differently (*Ootsuka et al., 2009*). One noticeable example of this is

that body temperature does not increase during REM sleep (*Alföldi et al., 1990*). Moreover, during extreme sleep disturbance (*Obermeyer et al., 1991*; *Baud et al., 2013*) or thermal manipulations (*Donhoffer et al., 1959*; *Kiley et al., 1984*), brain and body temperatures can deviate considerably, suggesting that their underlying regulatory mechanisms differ. Therefore, the extent of the model's accuracy in predicting changes in body temperature at small time scales should be tested experimentally.

## Implications

As the model can predict brain temperature dynamics with high accuracy even without optimization of the parameters in individual mice, a number of applications can be envisioned. For instance, neuroactive drugs do not only affect brain temperature (*Kiyatkin, 2018*) but often also impact the sleep–wake state. By predicting the effects of the sleep–wake state on brain temperature, the model can isolate the direct effect of the intervention on brain temperature, that is, the residuals of the simulation that cannot be explained by the sleep–wake state. Second, brain temperature affects many properties of neuronal functioning (*Kiyatkin, 2010*) and therefore may influence cognitive performance (*Walter and Carraretto, 2016*). Given the dynamics of brain temperature formulated by the model, it would be interesting to examine the putative associations between impaired cognitive functioning and distinctive thermoregulatory states, as has been observed during sleep inertia after awakening (*Kräuchi et al., 2004*). This would require determining first of all whether brain temperature in humans is governed by similar rules. Although such data in humans are sparse, one study reported similar temperature dynamics to those in the mouse (*Landolt et al., 1995*). The authors, however, reasoned that the underlying driving influence is circadian rather than the sleep–wake state, based on locomotor activity and NREM sleep depth arguments. Since studies in rodents have shown that both locomotor activity (*Shirey et al., 2015*; *Hoekstra et al., 2019*) and NREM sleep depth (*Franken et al., 1991*; *Tobler et al., 1994*) only have a minimal association with brain temperature, brain temperature in humans might be driven by the sleep–wake distribution, consistent with the assumptions of our simulation.

# Materials and methods

## Data acquisition

Detailed descriptions of the data acquisition, surgical procedures, and experimental design can be found elsewhere (*Mang and Franken, 2012*; *Hoekstra et al., 2019*) but is briefly described in this section. Data from 11 male C57BL6/J mice (seven wild types [WT]) and four lacking the gene encoding cold-inducible RNA-binding protein (*Cirbp* KO mice), 10–15 weeks of age were included. Sleep–wake distribution and brain temperature were unaltered in KO mice (*Hoekstra et al., 2019*) and likewise the results reported here did not differ statistically between the two genotypes. All mice were housed individually under a 12:12 hr light–dark cycle with ZT0 and ZT12, corresponding to light onset and dark onset, respectively. Ambient temperature was maintained at 25°C and food and water were provided ad libitum. Electroencephalograms (EEGs), recorded from a frontal-parietal derivation, and electromyograms (EMGs), recorded from the neck muscles, were used to 'score' the sleep–wake states 'wakefulness', 'NREM sleep', and 'REM sleep', at a 4 s resolution. Sleep–wake states marked as having EEG artefacts were included in the temperature analyses. Brain temperature was measured by a thermistor placed on top of the right visual cortex corresponding to the midpoint of the frontal-parietal EEG electrode pair on the left hemisphere, and was sampled at 10 Hz and the median value for each 4 s epoch represented that epoch. Recordings lasted 96 hr, started at light onset, and included two 24 hr days serving as baseline (termed baseline 1 and baseline 2), 6 hr of SD starting at light onset on day 3, followed by an 18-hr (recovery 1) and a 24-hr (recovery 2) recovery period. SD was achieved by gentle handling (*Mang and Franken, 2012*). In addition to the main experiment above, a subset of five animals (1 WT and 4 KO mice) was subjected to a 2- and 4-hr SD following the same protocol as for the 6-hr SD experiment but without recovery 2 (*Hubbard et al., 2020*). One session was excluded due to a technical problem resulting in abrupt discontinuity of the temperature data. A final, independent cohort of five WT mice of the same strain, same sex, same age, and that underwent the same 4-day protocol with 6 hr SD, was used to

test the predictions of the model. All experiments were approved by the Ethics Committee of the State of Vaud Veterinary Office Switzerland under license VD2743 and VD3201.

## Analysis

### Model 0: optimization details

All code was programmed in Matlab and optimization used the *fmincon* function of the Optimization Toolbox, by minimizing the mean squared error between the simulated and recorded temperature signals. All free parameters were always optimized simultaneously, even when additional ones were added at later stages. For each animal, we constrained the base values of both asymptotes to the range of empirical temperature values plus a 2°C deviation in either direction. The upper and lower asymptotes were always defined as vectors with same size (86'400 4 s epochs) as the 96 hr experiment recordings. The code assumed 12:12 hr light–dark cycles, the start of recording at light onset, sleep scoring only included wake, NREM sleep and REM sleep states, and 48 hr of baseline.

### Model 1: modulation of asymptotes according to prior wake-prevalence

To modulate the asymptotes at each time point based on the preceding prevalence of the sleep–wake state (referred to as the 'prior wake-prevalence'), for each time $t$, we calculated the fraction (0–1) of time spent in wake or REM sleep within a given time window ending at time $t$–1. We estimated temperature values for time points prior to the start of the recording (i.e., before the light onset of baseline 1), by averaging corresponding time points from the two days of baseline. Finally, we subtracted the averaged wake/REM fraction of total recording time (4 days) and multiplied by 2 so that the resulting values vary between − 1 and +1, and then multiplied the result by a scale factor to translate the fraction of time spent awake/REM sleep into a degree Celsius modulation of the asymptote. In addition, to enable the window not to end strictly at time $t$–1, we also implemented a window shift relative to time $t$ by moving the produced vector back and forth in time. Since this shift was allowed in both directions and we could not predict values after the end of the recording, we assumed zero values (i.e., no modulation of the asymptotes). The outcome vector was added to both asymptotes.

In contrast to the window scale factor and all other free parameters in this study, which were continuous variables, the window size and window shift were discrete parameters that eventually translated into integer numbers of specific cell indices in a vector. Due to the requirement of the *fmincon* function for differentiability, to optimize these two parameters we applied a brute force method: (1) we defined possible values for each of the two variables, (2) for each unique combination of the values we ran the *fmincon* on the rest of the parameters and calculated the error, and (3) kept the parameters (continuous and discrete) that yielded the lowest error output. In this study we chose to test window size values between 0 and 10 hr (step size of 0.25 hr) and window shift values between −5.0 and +0.5 hr (increments of 0.1 hr). For none of the mice was the best fit obtained with parameter values at the limits of these defined ranges.

### Model 2: addition of circadian modulation of the asymptotes

To introduce a circadian modulation of the asymptotes we used the following formulas:

$$L_t = L_t - \sin(2 \cdot \pi \cdot t + P) \cdot A$$

$$U_t = U_t - \sin(2 \cdot \pi \cdot t + P) \cdot A$$

where $A$ and $P$ are the free parameters for optimization and stand for the amplitude and phase (in hours) of the sine wave, respectively. $L_t$ and $U_t$ are the lower and upper asymptote values at time $t$; hence, both asymptotes were changed in parallel and to the same extent. The minus sign before the sine function is due to the fact that the recording started at light onset.

### Prediction of initial simulated temperature value without actual temperature data

In the original algorithm, the initial temperature was determined based on the recorded data, but to generalize our algorithm to predict brain temperatures of datasets without brain temperature

recordings, we needed a different method to estimate the value of the initial temperature. Although the temperatures of most animals ranged between 34°C and 36°C, some recordings showed non-physiological lower (32–34°C) or higher (36–38°C) ranges (*Hoekstra et al., 2019*), which probably resulted from technical problems. Nevertheless, the difference between the asymptote values was stable across animals (around 2°C), and simulation performance was high, regardless of the average absolute levels (*Table 1*). Therefore, to estimate the initial temperature when exclusively using the state sequence, we produced a predictive formula based on our existing data. First, we normalized the temperature data of each recording in the 96 hr experiment to a range between 34°C and 36°C. Then, we calculated the correlation between the percentage 'occurrence' of wake/REM sleep in a window at the start of the recording, and the average temperature in the same time window. In this way, we could reliably predict the temperature in the first minutes of the recording (r = 0.98, *Figure 4—figure supplement 1*), which we used as an estimated initial temperature. We chose a 7 min window size since it yielded the highest correlations across values between 1 and 10 min (analysis not shown). Finally, we applied linear regression to obtain the following equation:

$$\text{initial temperature} = 0.92265 \cdot \text{occurrence} + 34.3282$$

where *occurrence* is a value between 0 and 1, with 0 referring to 7 min of continuous NREM sleep and 1 to continuous REM sleep and/or waking. Note that this estimate of the initial temperature is valid solely for recordings that start at light onset and under a 12:12 hr light–dark cycle.

## Units and statistics

Throughout this manuscript, the time measures are expressed in hours, the temperature in degrees Celsius, and correlation coefficients (r) are the outcomes of Pearson correlations, unless stated otherwise. Optimized parameters are summarized as median values, and correlation coefficients as averages after Fisher transformation (*Fisher, 1915*; *Silver and Dunlap, 1987*). Paired Student's *t*-tests with false discovery rates correction (*Benjamini and Hochberg, 1995*) for multiple comparisons at a significance level of p<0.05 tested for significant effects for the 96 hr time series. Individual fits are summarized as the median of the root mean square (RMS) of the difference between the simulated data and the recorded data (referred to as 'RMS error' in the text), while effect sizes of the model residuals were assessed by the RMS of the mean residuals for hourly values (referred to as the 'residual RMS'). Differences in parameters across genotypes and models were assessed by *t*-tests, or by rANOVA with Tukey's range test when more than two values were assessed. The relative contribution of the three factors (sleep–wake state, prior wake prevalence, and circadian) to the variance explained by Model 2 was calculated as follows. The model output was first disassembled into three traces, corresponding to the effects of sleep–wake state, prior wake prevalence, and circadian. The unique variance explained by each factor was then calculated as the ratio of the variance of each factor's trace to the overall variance in the data. For the shared variance among factors, we subtracted the corresponding unique explained variances from the variance of the sum of the traces. The results of explained variance are presented as a Venn diagram using a tool available online (*Micallef and Rodgers, 2014*).

## Code availability

The two core Matlab scripts for brain temperature simulation (Source Code File 1) and parameters optimization (Source Code File 2) are provided as Supplementary material, together with running scripts (Source Code Files 3 and 4) and an example of the data (Source Code File 5). In addition, the recorded temperatures, sleep scoring, and simulated temperatures of the three models are available for each of the 11 animals in the main experiment, as an Excel file (*Supplementary file 4*).

## Acknowledgements

This study was supported by the Azrieli Foundation (Azrieli Fellowship Award supporting YS), the Swiss National Science Foundation (SNF n°146694 to PF supporting MMBH), and the State of Vaud (supporting YS, MMBH, and PF).

## Additional information

### Funding

| Funder | Grant reference number | Author |
|---|---|---|
| Azrieli Foundation | Azrieli Fellowship Award | Yaniv Sela |
| Swiss National Science Foundation | SNF n°146694 | Paul Franken |
| State of Vaud | | Yaniv Sela<br>Marieke MB Hoekstra<br>Paul Franken |

The funders had no role in study design, data collection and interpretation, or the decision to submit the work for publication.

### Author contributions

Yaniv Sela, Software, Formal analysis, Investigation, Visualization, Writing - original draft, Writing - review and editing; Marieke MB Hoekstra, Conceptualization, Investigation, Methodology, Writing - review and editing; Paul Franken, Conceptualization, Resources, Supervision, Funding acquisition, Writing - original draft, Writing - review and editing

### Author ORCIDs

Yaniv Sela (ID) https://orcid.org/0000-0002-2244-2067
Marieke MB Hoekstra (ID) http://orcid.org/0000-0003-0723-2026
Paul Franken (ID) https://orcid.org/0000-0002-2500-2921

### Ethics

Animal experimentation: All experiments were approved by the Ethicsal Committee of the State of Vaud Veterinary Office Switzerland under license VD2743 and VD3201.

### Decision letter and Author response

Decision letter https://doi.org/10.7554/eLife.62073.sa1
Author response https://doi.org/10.7554/eLife.62073.sa2

## Additional files

### Supplementary files

- Source code 1. A Matlab code for the brain temperature simulation based on sleep scoring data.

- Source code 2. A Matlab code for optimization of the parameters needed for simulation of brain temperature, by minimizing the mean squared error of the difference between the model and the temperature recorded.

- Source code 3. A Matlab code of an example for simulating brain temperature using Source Code File 1. The code uses the data example in Source Code File 5.

- Source code 4. A Matlab code of an example for parameters optimization using Source Code File 2. The code uses the data example in Source Code File 5.

- Source code 5. A Matlab file that includes example data (recorded temperature and respective sleep scoring) from one mouse, for running the example code files (Source Code Files 3 and 4).

- Supplementary file 1. Model 0 parameters for each animal. A table showing the optimized values for each of the four parameters of Model 0 without modulation of asymptotes, and the three additional descriptive variables, as in *Table 1*.

- Supplementary file 2. Model 1 parameters for each animal. A table showing the optimized values for each of the parameters of the model after introducing a modulation of both asymptotes

according to the prior wake-prevalence in the window preceding the assessment of temperature. Further details as in *Table 1*.

- Supplementary file 3. Performance of Model 2 for additional sleep deprivation (SD) experiments of the same animals. The table shows the Pearson's correlation coefficient (r) and root mean squared (RMS) error for five animals from the main experiment, after undergoing additional SD of shorter duration. Due to technical problems, the 2 hr SD experiment is missing for mouse number 622. See *Table 1* for the individual optimized parameters used (asterisks denote KO mice).

- Supplementary file 4. An Excel file documenting the recorded and simulated data for all 11 animals of the main experiment, in 4 s resolution. The data include the sleep scoring, the recorded temperature, and the simulated temperature for each of the three models.

- Transparent reporting form

### Data availability

The two core Matlab scripts for brain temperature simulation and parameters optimization, are provided as supplementary material, together with an example of the data and running scripts. In addition, the recorded temperatures, sleep scoring, and simulated temperatures of the three models are available for each of the 11 animals in the main experiment, as an Excel file.

The following datasets were generated:

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
