## [Decision Letter]

Thank you for submitting your article "A Sub-Minute Resolution Prediction of Brain Temperature Based on Sleep-Wake State in the Mouse" for consideration by *eLife*. Your article has been reviewed by three peer reviewers, one of whom is a member of our Board of Reviewing Editors, and the evaluation has been overseen by Catherine Dulac as the Senior Editor. The reviewers have opted to remain anonymous.

The reviewers have discussed the reviews with one another and the Reviewing Editor has drafted this decision to help you prepare a revised submission.

Summary:

Brain temperature is of neurobiological and clinical importance. The authors previously used a statistical approach to demonstrate that hourly brain-temperature values strongly co-varied with time-spent-awake, separate from locomotion. they now have developed a mathematical tool to simulate and predict mice cortical temperature based on the 4-second sleep-wake sequence. The model estimated temperature precisely with 91% of its variance based on three main factors: sleep-wake sequence, time-of-day (“circadian”), and a novel “prior wake prevalence”. With similar accuracy the model predicted temperature in a second, independent cohort using the parameters optimized for the first.

This manuscript tackles an important question of wake-sleep-dependent changes in brain temperature. The conventional view is that body temperature in mice is regulated by the circadian clock and it still often comes as a surprise to many that in fact the observed differences between day and night are primarily driven by the state of vigilance. The relationship between wakefulness and increased body temperature (or perhaps more correctly the relationship between sleep and hypothermia) has been established before, but the study of Sela et al. goes to great length to quantitatively describe the temperature dynamics. The authors conclude that model "can help differentiate thermoregulatory from sleep-wake driven effects".

Essential revisions:

1) First, how vigilance state and sleep-wake history affect the relationship between the brain and body temperature remains unclear. Very few laboratories monitor brain temperature routinely, and the circadian field relies almost entirely on skin or core body temperature recording. Clearly, the authors do not have access to such data in the data set they used, but it is essential to be discussed. Are they independent? If the time constants were different, it could provide important insights into the underlying mechanisms and also functional significance.

2) To this end, while the effects of vigilance states on temperature dynamics are nicely documented, the underlying causes or biological significance of the effects observed remain unclear. The authors mention that "Brain temperature affects many properties of neuronal functioning", but discussing some specific examples, illustrating which properties are actually influenced by changes within the temperature range investigated here, may help. Conversely, I am still not entirely clear what are the factors that actually drive the changes in brain temperature observed.

3) The observations made by the authors clearly apply to artificial laboratory conditions at a specific ambient temperature only, but it remains unclear how the vigilance state affects body/brain temperature at different ambient temperatures. Would you still observe vigilance-state specific changes in brain temperature if mice were kept at thermoneutrality? The authors optimise some specific parameters, such as "window length", "window shift" or "scaling factor", which helps to obtain a better fit, but then the question remains whether these would also work at different ambient temperatures, and what is their biological meaning.

4) Arguably, the actual temperature in the nest is higher than ambient temperature. Is the drop in temperature during sleep smaller when the animal sleeps in the nest?

5) Finally, what is the relationship between locomotor activity and brain temperature? Does it matter if the animal is involved in an intense exploratory or running behaviour vs. relatively quiet wakefulness? Would the model work just as well in both cases?

---

## [Author Response]

Essential revisions:1) First, how vigilance state and sleep-wake history affect the relationship between the brain and body temperature remains unclear. Very few laboratories monitor brain temperature routinely, and the circadian field relies almost entirely on skin or core body temperature recording. Clearly, the authors do not have access to such data in the data set they used, but it is essential to be discussed. Are they independent? If the time constants were different, it could provide important insights into the underlying mechanisms and also functional significance.

Thank you for raising this important issue. Important also because many labs indeed rely on body temperature measurements. We therefore have added a section to the Discussion addressing this matter (“Brain versus body temperature”).

The general time course of body and brain temperature changes in parallel (Blessing, 2018) suggesting their dynamics underlie similar rules such that sleep-wake state explains an important part of the variance in body temperature as well. We therefore anticipate that the daily dynamics of e.g. hourly mean values of body-temperature can be predicted using similar model assumptions, which could already be highly useful for circadian-related research questions. However, at the finer time-scales at which we optimized our model, the temperature of the brain and body deviate (Ootsuka et al., 2009) and e.g. body temperature does not increase during REM sleep (Alföldi et al., 1990). We are therefore not convinced that the more rapid changes in brain states will be able to predict body temperature changes and, vice versa, that other factors such as locomotion will influence body temperature much more than it does brain temperature (see point 5 below).

2) To this end, while the effects of vigilance states on temperature dynamics are nicely documented, the underlying causes or biological significance of the effects observed remain unclear. The authors mention that "Brain temperature affects many properties of neuronal functioning", but discussing some specific examples, illustrating which properties are actually influenced by changes within the temperature range investigated here, may help. Conversely, I am still not entirely clear what are the factors that actually drive the changes in brain temperature observed.

In the first paragraph of the Introduction, where we mentioned the biological significance of central temperature variation on brain function, we now have added specific examples as suggested. We also expanded the second paragraph of the Introduction to explain the factors thought to determine brain temperature.

Measured brain temperature is considered to result from the balance between heat production (brain metabolism) and heat dissipation (brain blood flow and the brain-to-blood temperature gradient; see revised second paragraph of the Introduction). Changes in sleep-wake state affect both heat production and dissipation: metabolism and oxygen consumption are increased during wakefulness and REM sleep relative to NREM sleep (Nir et al., 2013), blood flow to the brain is relatively increased (after correcting for the large drop in oxygen consumption during NREM sleep; McAvoy et al., 2019), and, as body temperature is actively down-regulated through peripheral vasodilation and perspiration at NREM sleep onset, also the brain-to-blood temperature gradient increases (Szymusiak, 2018). Accordingly, the simulation identified sleep-wake transitions to be the main driver of the observed changes in brain temperature over the 96h experiment, with as additional factors “prior wake-prevalence” and “circadian time”. As these 3 factors combined explain over 91% of the variance in brain temperature, we consider these to be main contributing factors.

3) The observations made by the authors clearly apply to artificial laboratory conditions at a specific ambient temperature only, but it remains unclear how the vigilance state affects body/brain temperature at different ambient temperatures. Would you still observe vigilance-state specific changes in brain temperature if mice were kept at thermoneutrality? The authors optimise some specific parameters, such as "window length", "window shift" or "scaling factor", which helps to obtain a better fit, but then the question remains whether these would also work at different ambient temperatures, and what is their biological meaning.

Differences in ambient temperature modulate the efficacy with which the body dissipates heat to the environment, which, in turn, affects the temperature of the blood entering the brain (Hayward and Baker, 1969). Moreover, variations in ambient temperature strongly influence sleep-wake behavior (Kräuchi and Deboer, Front Biosci 2010). Nevertheless, brain-temperature dynamics at sleep-wake transitions remain surprisingly similar over a broad range of ambient temperatures (10-29°C), although the increase at the highest temperature seemed smaller, possibly related to the higher absolute cortical temperatures reached at this ambient temperature (Alfoldi et al., 1990). The parameters might therefore need to be adjusted according to ambient temperature so that predicted values will optimally fit. Since the basic dynamic of brain temperature between states is qualitatively preserved across different ambient temperatures, we estimate that the factors we incorporated in the model, including the parameters related to “prior wake prevalence” that the reviewer refers to, will be the same, although the parameter values might need to be re-optimized. We have now discussed these issues in the Discussion under the new section “Additional factors affecting brain temperature dynamics”. The biological significance of the “prior wake prevalence” factor was already discussed under the section “Model parameters” and has been elaborated on in the revised version.

4) Arguably, the actual temperature in the nest is higher than ambient temperature. Is the drop in temperature during sleep smaller when the animal sleeps in the nest?

Nesting helps to keep the animal in a more comfortable thermal zone with less energy expenditure (Harding et al. Curr Opin Physiol 2020) and therefore “nest quality” decreases at higher ambient temperatures (Gordon, 2017). As discussed above, ambient temperature might produce quantitatively different dynamics and therefore the parameters will need to be re-optimized to take into account whether the animal sleeps inside or outside the nest. An equally important factor is whether mice sleep alone or in groups. We now have discussed these issues in the Discussion under the new section “Additional factors affecting brain temperature dynamics”.

5) Finally, what is the relationship between locomotor activity and brain temperature? Does it matter if the animal is involved in an intense exploratory or running behaviour vs. relatively quiet wakefulness? Would the model work just as well in both cases?

It has been hypothesized that brain temperature is primarily determined by heat generated in skeletal muscles, which is carried from the periphery to the central nervous system by the bloodstream (Moser and Mathiesen, Neuroreport 1996). However, other studies reported very low, or even no, correlation of core body temperature with locomotor activity (Refinetti, Physiol Behav 1994, Weinert et al. Physiol Behav 1998, Hilmer et al. J Thermal Biol 2010). More specifically, in our previous work we directly examined the contribution of locomotor activity on brain temperature (Hoekstra et al., 2019) and found its contribution to be very modest (explaining ca. 2% of variance in brain temperature), in line with other studies (Shirey et al., 2015).

In the current experiment mice did not have access to running wheels and did not engage in intense locomotor activity maybe explaining why home cage locomotor activity does not notably contribute to brain temperature. Intense locomotor behavior may indeed modulate brain temperature. However, to capture also such changes in the model, wakefulness can no longer be treated as a unique state and an additional factor will have to be quantified and added to the simulation. We have discussed these issues in the Discussion under the new section “Additional factors affecting brain temperature dynamics”.